# Phenotypic screening of 1,953 FDA-approved drugs reveals 26 hits with potential for repurposing for Peyronie's disease

**Marcus M. Ilg[1], Alice R. Lapthorn[1], David J. Ralph[1,2], Selim Cellek[1]***

**1** Medical Technology Research Centre, School of Allied Health, Faculty of Health, Education, Medicine and Social Care, Anglia Ruskin University, Chelmsford, United Kingdom, **2** Department of Urology, University College London Hospital, London, United Kingdom

* selim.cellek@aru.ac.uk

**Data Availability Statement:** The data underlying the results presented in the study are available from figshare at https://doi.org/10.25411/aru.21501531.v1.

## Abstract

Drug repurposing has been shown to bring safe medications to new patient populations, as recently evidenced by the COVID-19 pandemic. We investigated whether we could use phenotypic screening to repurpose drugs for the treatment of Peyronie's disease (PD). PD is a fibrotic disease characterised by continued myofibroblast presence and activity leading to formation of a plaque in the penile tunica albuginea (TA) that can cause pain during erection, erectile dysfunction, and penile deformity. PD affects 3–9% of men with treatment options limited to surgery or injection of collagenase which can only be utilised at late stages after the plaque is formed. Currently there are no approved medications that can be offered to patients presenting with early disease before the formation of the plaque. Drug repurposing may therefore be the ideal strategy to identify medical treatments to address this unmet medical need in early PD. We used primary human fibroblasts from PD patients in a phenotypic screening assay that measures TGF-β1-induced myofibroblast transformation which is the main cellular phenotype that drives the pathology in early PD. A library of FDA-approved 1,953 drugs was screened in duplicate wells at a single concentration (10 μM) in presence of TGF-β1. The myofibroblast marker α-SMA was quantified after 72h incubation. A positive control of SB-505124 (TGF-β1 receptor antagonist) was included on each plate. Hits were defined as showing >80% inhibition, whilst retaining >80% cell viability. 26 hits (1.3%) were identified which were divided into the following main groups: anti-cancer drugs, anti-inflammation, neurology, endocrinology, and imaging agents. Five of the top-ten drugs that increase myofibroblast-transformation appear to act on VEGFR. This is the first phenotypic screening of FDA-approved drugs for PD and our results suggest that it is a viable method to predict drugs with potential for repurposing to treat early PD.

## Introduction

Recently, Covid-19 has shone a new light on drug repurposing as the world needed to quickly discover effective treatments for Covid-19 patients during the pandemic and found success in repositioning of tocilizumab (a drug to treat arthritis) and dexamethasone (a corticosteroid)

**Funding:** The authors received no specific funding for this work.

**Competing interests:** The authors have declared that no competing interests exist.

[1, 2]. Compared to the development of new chemical entities, repurposed drugs have the potential to reach patients faster, as they are generally approved sooner (12–18 months) at a 50–60% reduced cost [3, 4]. The field of urology itself is no stranger to drug repurposing, as evidenced by the use of phosphodiesterase type 5 inhibitor sildenafil for erectile dysfunction. Sildenafil was previously developed for treatment of angina and gained a novel indication through observational studies and serendipity [5]. Repurposing often happens through off-label prescription, particularly in the US where 1 in 5 prescriptions are for off-label use, according to the Agency for Healthcare Research and Quality [6]. The vast amount of data available for 'old' drugs make it appealing to repurpose them for other indications, as US Food and Drug Administration (FDA) and European Medicines Agency (EMA) have previously evaluated these drugs thoroughly. In fact, the EMA is now starting a pilot to support academic efforts to repurpose authorised medicines for new indications [7]. In England, the National Health Service (NHS) has recently released a report of the Medicines Repurposing Pro-gramme, making recommendation on opportunities to repurpose drugs to improve outcomes, patient experience, equity of access and value for money [8].

Peyronie's disease (PD) is characterised by a fibrotic plaque in the penile tunica albuginea (TA), that can lead to pain, curvature, and erectile dysfunction [9]. PD can be divided into an acute (early) phase (usually within the first 6 months after the first symptoms, pain during erection and appearance and gradual worsening of the curvature) and a chronic (late) phase (after ~12 months, stable plaque and curvature) [10]. Despite affecting 3–9% of men, current treatment is limited to surgery as non-surgical treatment has demonstrated limited efficacy [10, 11]. Currently, the only non-surgical therapy remains the intralesional injection of colla-genase [12], which has been withdrawn from the market recently [13]. Both surgery and colla-genase are used as a last resort at late stages of the disease. Therefore, there are no approved medications that can be offered to patients representing with early disease, typically before for-mation of a plaque.

The etiological mechanism suggested for PD is repetitive trauma to the tunica albuginea [14], resulting in fibrinogen extravasation which in turn leads to expression of TGF-β1 [15]. Overexpression of TGF-β1 is considered a key factor in PD [16, 17] and other fibrotic diseases [18].

A key player in PD and other fibrotic diseases is the myofibroblast, a contractile, highly pro-liferative, extracellular matrix producing, alpha-smooth muscle actin (α-SMA)-expressing cell type. Transformation of resident fibroblasts to myofibroblasts in response to inflammatory cyto-kines such as TGF-β1 has been suggested to be the main cellular phenotype that drives fibrosis [15, 19–21]. Targeting myofibroblasts by either inhibiting their transformation [22] or selec-tively eliminating them [23] has been suggested as therapeutic approach for fibrosis [22–24].

Although fibrotic diseases account for 45% of mortality in the Western world [25], the available approved treatment options are limited to two drugs (pirfenidone and nintedanib), both of which are indicated only for pulmonary fibrosis despite years of research and invest-ment in the fibrosis field [26]. One reason for lack of progress might be the over-reliance on single target-based drug discovery, which is hindered drastically by the redundant pathways and compensatory mechanisms typically present in fibrosis [25]. Phenotypic drug discovery alleviates some of these issues, as phenotypic screening assays are target-agnostic and do not require a hypothesis for the role of the target in the disease pathophysiology [27]. The interest in phenotypic assays has been catalysed by the landmark paper by Swinney in 2011 that detailed how phenotypic screening assays outperformed the target-based approach in first-in-class drug discovery [28]. Whilst phenotypic assays have been described to recapitulate the dis-ease biology in a more relevant fashion [29], there is also a drive to utilize the benefits of phe-notypic screening assays to advance drug repurposing efforts [30].

We have developed a phenotypic assay that can reproducibly measure the transformation of human primary fibroblasts obtained from patients with PD to myofibroblasts [24, 31, 32]. Using this assay, we had previously evaluated 21 FDA-approved drugs and identified two classes, phosphodiesterase type 5 inhibitors (PDE5is such as sildenafil, vardenafil and tadalafil) and selective estrogen receptor modulators (SERMs such as tamoxifen and raloxifene), that synergised in *in vitro* and *in vivo* models of PD [24]. In this study we expanded our screening to 1,953 FDA-approved drugs. We utilized an FDA-approved drug library to make use of the already known safety and tolerability data and thereby shorten potential clinical trials of hit drugs. The complete dataset for all 1,953 drugs is provided in this manuscript to help drive further repurposing efforts for PD.

## Material and methods

### Ethics, tissue acquisition, and isolation of fibroblasts

Samples were acquired as previously described [24, 31, 32]. TA tissue samples were collected from patients undergoing surgery for PD at University College London Hospital (UCLH), London, UK. All patients gave fully informed written consent to the study. This study was approved by independent research ethics committees (NRES Committee East of England 12/EE/0170 and NRES Committee North of Scotland 15/NS/0051). Non-plaque TA was obtained from PD patients undergoing a Nesbit procedure whereby non-fibrotic TA tissue was excised from the opposite side of the plaque and the penis was straightened but shortened. Tissue samples were carefully dissected to ensure that all cavernosal tissue was removed from the TA. Fibroblast cultures were established by dissecting TA fragments and anchoring them into six-well tissue culture plates (Fisher Scientific, Loughborough, United Kingdom) as described previously [24, 31]. The tissue pieces were incubated in DMEM-F12 (GIBCO, Invitrogen, Waltham, Massachusetts, United States) containing 10% FCS (Fisher Scientific) and 1% penicillin-streptomycin (GIBCO, Invitrogen) at 37˚C and 5% $CO_2$, for 5–7 d. When cell outgrowth was observed, tissue fragments were removed. Cells were sub-cultured when confluent and passages 1–6 were used for all experiments. Before commencing experiments, fibroblast identity of cells was confirmed as previously described [24]. Briefly, cells were considered fibroblasts if they expressed vimentin in the absence of desmin and started expressing α-SMA after treatment with TGF-ß1. Cells derived from one patient (58 years old, no co-morbidities, no co-medication) were used for the drug screening campaign. Cells derived from two other patients (58 years old with sleep apnoea, no co-medication; 33 years old with hypertension and right inguinal hernia, received ramipril) were used to confirm hits in concentration response curves.

### Screening assay

The media containing TGF-ß1 and/or the FDA-approved drugs was prepared and dispensed by an automated liquid handling system (PipetMAX, Gilson, UK). A commercially available library of 1,953 FDA-approved drugs (Selleck Chemicals, USA) was tested. Briefly, 10 mM stock drugs were diluted 1:100 in DMEM/F12 basal medium to generate a stamp plate, which was subsequently diluted 1:10 in full cell culture media (DMEM/F12 supplemented with 10% foetal bovine serum (FBS) and 1% penicillin-streptomycin) containing 10 ng/mL TGF-ß1 (Sigma-Aldrich, Gillingham, UK) to generate a 10 μM solution. Drugs were tested in duplicate wells and every plate included the following controls: untreated, 10 ng/mL TGF-ß-1 only, and 10 ng/mL TGF-ß1 in presence of 1 μM SB505124 (a TGFBRI inhibitor) as positive control. Plates were stained using the In-Cell ELISA protocol, outlined below. Data was analysed as described below. Integrity of the library was confirmed by ordering three hit compounds from a different supplier (Sigma-Aldrich, UK) and constructing full concentration-response-curves with them.

## In-Cell ELISA

In-Cell ELISA (ICE) was performed as previously described [24, 31, 32]. Briefly, fibroblasts were seeded onto 96-well, optical, flat-bottom, black microplates (Nunc, Fisher Scientific) at $5x\,10^3$ cells/well. After overnight attachment, they were incubated with or without 10 ng/ml TGF-ß1 in presence or absence of additional compounds for 72 h. After the incubation, the cells were fixed with 4% paraformaldehyde, washed and blocked with 10% donkey serum and 0.1% Triton X-100 in phosphate-buffered saline (PBS). This was followed by primary antibody incubation using an anti–a-SMA antibody (1:3000; Sigma-Aldrich, Gillingham, United Kingdom) for 2 h. After washing steps, the cells were incubated with donkey anti-mouse secondary antibody conjugated to an infrared dye that emits at 800 nm (1:500; IRdye 800CW; Li-COR, Cambridge, United Kingdom) and a nuclear counterstain at that emits at 700 nm (1:1000; DRAQ5; Biostatus, Loughborough, United Kingdom) for 1 h. The plate was scanned using an infrared imaging system (Odyssey CLx imager; LI-COR) at both 700 and 800 nm wavelengths.

## Immunocytochemistry

Immunocytochemistry (ICC) was performed as previously described [24]. Briefly, cells were seeded at a density of $5x10^4$ cells/well onto sterile glass cover slips in 6-well plates. After overnight attachment, cells were treated with TGF-ß1 and/or drugs or control conditions. After 72 h, the cells were fixed using ice-cold methanol, washed in PBS, and blocked using 10% donkey serum in PBS for 2 h. This was followed by primary antibody incubation using an anti–α-SMA antibody (1:1000; Sigma-Aldrich, Gillingham, United Kingdom) for 2 h. Cover slips were washed three times using PBS and cells were incubated with secondary antibody in PBS (1:1000; Abcam, Cambridge, United Kingdom) for 2 h. After washing, coverslips were mounted using VECTASHIELD® Antifade Mounting Media with DAPI (Vector Laboratories, Newark, California, United States). Images were captured using a Olympus microscope with a Leica DFC3000 G camera.

## Reverse transcriptase quantitative polymerase chain reaction

Cells were seeded at $5.0 \times 10^4$ cells/well into 6-well plates and incubated in presence or absence of 10 ng/ml TGF-ß1 with or without drugs (terfenadine, ebastine, and solifenacin) at 10 μM for 72h. Total RNA was extracted from the cells using RNeasy Mini Kit (Qiagen, UK) as per manufacturer's instructions. RNA was transcribed in 20-μl reactions using the UltraScript 2.0 cDNA Synthesis Kit (PCR BIOSYSTEMS, UK) utilising random priming and conditions specified by the manufacturer. A thermocycler with heated lid at 115˚C (G-Storm, UK) was used. The cDNA preparations were diluted 1:1 in RNase-free water for PCR reactions. The qPCR reactions were carried out in 5-μl reactions with 1x SensiFast SYBR master mixes (Bioline, UK). Primers were at a final concentration of 300 nmol and 1 μl of cDNA was used. The following programme was run of a CFX Connect (Bio-Rad, UK): enzyme activation at 95˚C for 3 minutes, followed by 40 cycles of 95˚C for 15 seconds, 60˚C for 30 seconds, and 72˚C for 30 seconds. Two reference genes, EIF4A2 and GAPDH, were used to normalize gene expression data.

## Western blot

Cells were seeded at $10 \times 10^4$ cells/well into 6-well plates and incubated in presence or absence of 10 ng/ml TGF-ß1 with or without terfenadine, ebastine, and solifenacin at 10 μM for 72h. Protein was isolated using RIPA buffer (Sigma Aldrich, UK). 20 μg of protein mixed with Li-Cor 4x protein loading buffer (Li-Cor, UK) and heat denatured at 95˚C under reducing conditions. Samples were then loaded onto Any kD• Mini-PROTEAN® TGX Precast Protein Gels

(Bio-Rad) along with a protein ladder (Bio-Rad). After gel electrophoresis, the separated protein was transferred onto a methanol-activated PVDF membrane (Millipore) by wet blotting at 350 mA for 1h. The membrane was washed and left to dry for 1h before being blocked with Li-Cor Intercept Blocking Buffer (Li-Cor) for 1h. The membranes were incubated overnight with primary antibodies (anti–α-SMA antibody, 1:3000 (Sigma-Aldrich); anti-fibronectin antibody, 1:1000 (ThermoFisher); anti-GAPDH, 1:10,000 (Abcam)) at 4˚C on a shaker. Membranes were washed 4x with Tris-buffered saline containing 0.1% Tween 20 before incubation with secondary antibodies (Li-Cor, 1:10,000) for 1h on a shaker in darkness. This was followed by 4 washes with Tris-buffered saline containing 0.1% Tween 20. Blots were visualised using an infrared imaging system (Odyssey CLx imager; LI-COR) at 700 and 800 nm wavelengths simultaneously.

## ECM production assay

ECM production assays were performed as previously described [24]. Briefly, fibroblasts were seeded onto 96-well, optical, flat-bottom, black microplates (Nunc, Fisher Scientific) at $5x\,10^3$ cells/well. After overnight attachment, they were incubated with or without 10 ng/ml TGF-ß1 in presence or absence of additional compounds for 7 days. Cells were incubated with DRAQ5 for 5 minutes and cell number was determined using an infrared imaging system (Odyssey CLx imager; LI-COR) at 700 nm. Cells were lysed using 0.25 M ammonium hydroxide and ECM was fixed with a solution containing 50% methanol and 7.5% acetic acid for 1 h at–20˚C. Afterwards, ECM was stained with Coomassie Blue overnight at 4˚C. Plates were scanned using an infrared imaging system (Odyssey CLx imager; LI-COR) at 700 nm.

## Data analysis

Protein expression data was normalised to nuclear staining intensity. Data variability of screening assay was previously assessed by use of Z' to assess assay signal dynamic range and the data variation associated with the signal measurements. The following formula that was used:

$$1 - \frac{3 - (\sigma p + \sigma n)}{(\mu p - \mu n)}$$

Where p refers to the positive controls (TGF-β1-treated) and n refers to the negative controls (untreated). Σp and σn are the standard deviations of the positive and negative controls, and μp and μn are the means of the positive and negative controls. Z' was consistently above 0.5, which proved the assay to be robust for high-throughput screening [24].

As the assay's variability and robustness were confirmed, percentage inhibition could be used as a metric to define a hit. To calculate the percentage inhibition, the following formula was used:

$$\frac{(\text{Positive control} - \text{Sample})}{(\text{Positive control} - \text{Negative control})} \text{ x } 100$$

Positive control refers to fibroblasts treated with TGF-β1 only media, negative control refers to fibroblasts given blank media. The mean of duplicate wells was used in these calculations. In some cases, the drugs elicited an effect strong enough to render the sample value below the low-level background expression of the negative control, which could result in inhibition above 100% when using the above formula. In these cases, the numerical value above 100% is considered full inhibition and is displayed as 100% inhibition, while negative inhibition means increased myofibroblast transformation. We considered a drug that exhibited 80% inhibition

while retaining 80% viability (80% of DRAQ5 expression of TGF-ß1 control) at 10 μM as a hit. Three hits were followed up by full concentration response curves in additional cell lines derived from other patients (total of three patients used, N = 3).

RT-qPCR results were normalized using the $2^{-\Delta\Delta CT}$ method to show fold-change of gene expression [33]. Three biological replicates were used, and data is presented as box & whisker plot to show biological variability. Statistical analysis for data sets was performed using Graph-Pad Prism 7 Software. Statistical significance was calculated using one-way analysis of variance. A p value of less than 0.5 was considered significant.

## Results

### Confirmation of the cell phenotype and the assay

We have previously published the data that confirmed the cell phenotype and validated the assay [24]. Briefly, using in-Cell ELISA (ICE) and immunocytochemistry we confirmed that the cells isolated from the TA were fibroblasts as they expressed vimentin but not desmin. As measured with expression of α-SMA using ICE, the fibroblasts transformed to myofibroblasts by TGF-ß1 in a concentration dependent manner with $EC_{50}$ value of 0.8 ng/ml. $EC_{80}$ of TGF-ß1 in the concentration response curve was 10 ng/ml which was used as the concentration for the screening campaign described below. The reproducibility of the assay was confirmed by achieving Z' of 0.89 which represents high reproducibility [34]. SB-505124 (TGF-β1 receptor antagonist [35]) was used to validate the assay which achieved an $IC_{50}$ value of 0.6 μM.

### Phenotypic screening campaign of 1,953 drugs

FDA-approved 1,953 drugs were tested at a single concentration of 10 μM, in duplicate wells with 30 drugs having been tested in a 96-well plate as shown in Fig 1A.

Fig 1B shows the percentage inhibition achieved with 1,953 drugs, which are also shown numerically in S1 Table. 170 drugs were shown to inhibit myofibroblast transformation by more than 80%, however, to exclude the possibility of cytotoxicity being the cause of myofibroblast inhibition, cell viability was assessed simultaneously. As a result of this, only 26 drugs (1.3% hit rate) fit the hit criteria, as they retained 80% viability (Table 1). We also observed that 106 drugs increased myofibroblast transformation, shown as negative inhibition (Fig 1B). The full list of 1,953 drugs along with their inhibition, cell viability, indication and other information, such as CAS number, target, DMSO (mg/mL) maximum solubility, DMSO (mM) maximum solubility, water (mg/mL) maximum solubility, water (mM) maximum solubility, URL, pathway, chemical formula, form, synonyms, SMILES (simplified molecular-input line-entry system), ALogP, and molecular weight are provided in S1 Table.

### Hits from the screening campaign

Table 1 lists the drugs that fit our hit criteria for both percentage inhibition and cell viability. The table also lists potential targets/mechanisms of the drugs and the distribution of drugs according to their indication. Most hits (10) are anti-cancer drugs, followed by anti-inflammation (7), neurology (3) and endocrinology (3), with 2 imaging agents. It appears that histamine signalling is a common target among the hits, being represented by 5 drugs.

Table 2 contains the top ten drugs that elicited the highest increase in myofibroblast transformation, indicated by negative percentage inhibition. 60% of these drugs are cancer drugs and it appears that vascular endothelial growth factor receptor (VEGFR) is a common target among these drugs. The drugs in this group are mainly kinase inhibitors, apart from semagacestat, a gamma-secretase inhibitor.

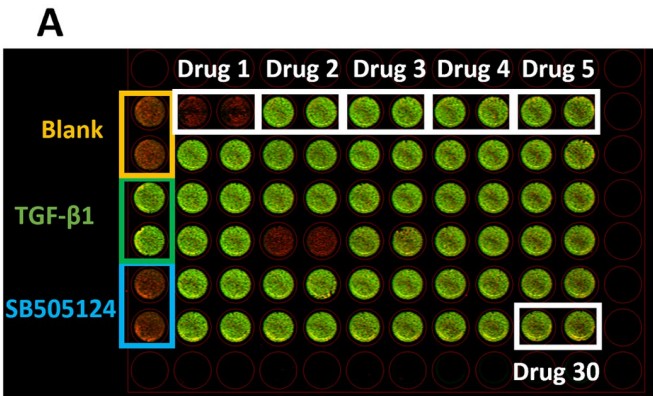

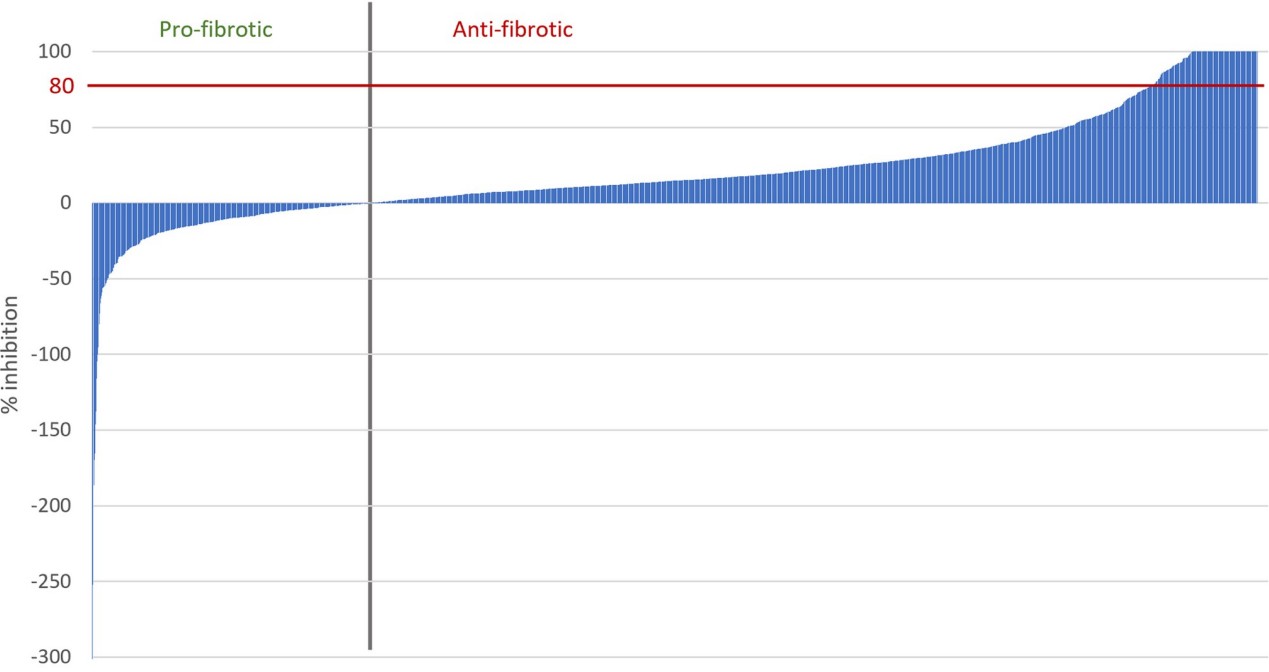

**Fig 1. Screening of 1,953 FDA-approved drugs using a phenotypic, high-throughput screening assay. A**: The image output for an example plate of the screening campaign. TA-derived fibroblasts were exposed to TGF-ß1 (10 ng/mL) and/or additional compounds/control conditions for 72h before being stained using ICE. First column depicts control wells in following order (top to bottom): two wells blank media (orange box), two wells TGF-ß1 only without any drugs (green box), two wells TGF-ß1 plus SB505124 (blue box). Other rows show duplicate wells (left to right) of TGF-ß1 plus 10 μM of a drug (white boxes showing six examples), with a total of 30 drugs per plate. For clarity purposes, only first 5 drugs and drug #30 are shown. Green colour indicates α-SMA expression, red colour indicates DRAQ5 nuclear stain. **B**: Graphical representation of percentage inhibition of myofibroblast transformation quantified for 1,953 FDA-approved drugs. Negative inhibition (left-hand side of grey vertical line) suggests increased myofibroblast transformation, positive inhibition (right-hand side of grey vertical line) suggests decreased myofibroblast transformation.

**Table 1. Hits from the screening campaign.** The table lists the 26 drugs in alphabetical order that met the hit criteria of >80% inhibition and >80% cell viability, also indicating their target/mechanism, and indication. Viability above 100% indicates increased cell proliferation.

| Drug Name | % inhibition | % viability | Target/mechanism | Indication |
|---|---|---|---|---|
| Aceclofenac | 92% | 86% | Cyclooxygenase (COX) inhibitor | Inflammation |
| Adenine HCl | 80% | 91% | DNA damage/RNA synthesis | Cancer |
| Alimemazine Tartrate | 87% | 92% | H1 histamine receptor antagonist | Inflammation |
| Amodiaquine dihydrochloride dihydrate | 100% | 84% | Histamine N-methyl transferase inhibitor | Inflammation |
| Chlorotrianisene | 95% | 83% | Non-steroidal oestrogen | Endocrinology |
| Dabrafenib (GSK2118436) | 88% | 136% | B-Rapidly Accelerated Fibrosarcoma (B-RAF) inhibitor | Cancer |
| Dabrafenib Mesylate | 91% | 127% | B-RAF inhibitor | Cancer |
| Dexamethasone Sodium Phosphate | 81% | 82% | Corticosteroid | Inflammation |
| Disodium Cromoglycate | 81% | 86% | Histamine release blocker | Inflammation |
| Dronedarone HCl | 100% | 84% | Multi ion channel blocker | Neurology |
| Ebastine | 100% | 101% | H3 histamine receptor antagonist | Inflammation |
| Entospletinib (GS-9973) | 85% | 105% | Spleen tyrosine kinase (Syk) inhibitor | Cancer |
| Gadopentetate Dimeglumine | 88% | 87% | MRI contrast agent | Imaging agent |
| Galunisertib (LY2157299) | 100% | 204% | TGFBRI inhibitor | Cancer |
| Iopromide | 86% | 90% | Contrast agent | Imaging agent |
| Lapatinib | 100% | 89% | EGFR and ErbB2 inhibitor | Cancer |
| Lypressin Acetate | 100% | 83% | Antidiuretic hormone | Endocrinology |
| Methylene Blue | 100% | 141% | Contrast agent, used for methemoglobinemia | Imaging agent |
| Mitoxantrone 2HCl | 100% | 90% | Topoisomerase type II inhibitor | Cancer |
| Molidustat (BAY 85–3934) | 86% | 81% | Hypoxia-inducible factor prolyl hydroxylase (HIF-PH) inhibitor | Endocrinology |
| Prochlorperazine Dimaleate | 92% | 80% | Dopamine (D2) receptor antagonist | Neurology |
| Rigosertib (ON-01910) | 95% | 80% | Inhibitor of polo-like kinase 1 (PLK1) | Cancer |
| Saracatinib (AZD0530) | 100% | 81% | Proto-oncogene tyrosine-protein kinase Src inhibitor | Cancer |
| Solifenacin succinate | 95% | 81% | Muscarinic receptor antagonist to treat overactive bladder | Neurology/urology |
| Terfenadine | 90% | 91% | H1 histamine receptor antagonist | Inflammation |
| Vincristine sulfate | 87% | 80% | Prevents tubulin polymerisation | Cancer |

**Table 2. Top ten drugs that increase myofibroblast transformation.** The table lists the 10 drugs in alphabetical order that increased myofibroblast transformation the most (negative % inhibition), also indicating cell viability, the target of the compound and the current indication.

| Drug Name | % inhibition | % viability | Target | Indication |
|---|---|---|---|---|
| Axitinib | -165% | 85% | Cluster of differentiation 117 (CD117), Platelet-derived growth factor (PDGFR), VEGFR | Cancer |
| Barasertib (AZD1152-HQPA) | -100% | 65% | Aurora Kinase | Cancer |
| Cabozantinib (XL184, BMS-907351) | -252% | 58% | Mesenchymal-epithelial transition factor (c-Met), VEGFR | Cancer |
| Lenvatinib (E7080) | -186% | 62% | VEGFR | Cancer |
| Nintedanib (BIBF 1120) | -105% | 77% | Fibroblast growth factor receptor (FGFR), PDGFR, VEGFR | Lung fibrosis |
| Oclacitinib maleate | -146% | 85% | Janus kinase (JAK) | Inflammation |
| Pimasertib (AS-703026) | -170% | 67% | Mitogen-activated protein kinase kinase (MEK) | Cancer |
| Semagacestat (LY450139) | -327% | 70% | Gamma-secretase | Neurological |
| Tivozanib (AV-951) | -116% | 69% | VEGFR | Cancer |
| Trametinib (GSK1120212) | -138% | 59% | MEK1, MEK2 | Cardiovascular Disease |

## Confirmation of hits

To confirm the integrity of the library and to ensure that hits have a concentration-dependent anti-myofibroblast effect *in vitro*, three hits, terfenadine, ebastine, and solifenacin were chosen randomly to follow up with full concentration-response curves. They were purchased from a different supplier and tested in concentrations ranging from 0.03 to 100 μM. Fig 2 represents the concentration-dependent effects of the drugs. An inverse sigmoid curve with upper and lower plateau could be generated for terfenadine (a histamine receptor-1 antagonist), with an $IC_{50}$ of 1.7 μM (Fig 2A). Cell viability was affected at concentration of 30 μM and higher. Fig 2B shows the concentration-dependent decrease in myofibroblast transformation in cells treated with various concentrations of ebastine (a histamine receptor-3 antagonist). An $IC_{50}$ of 4.3 μM could be calculated, with cell viability affected at a concentration of 100 μM only. An inverse sigmoid curve with upper and lower plateau could also be generated for the effect of solifenacin (a muscarinic receptor antagonist) on myofibroblast transformation (Fig 2C). An $IC_{50}$ of 3.5 μM was calculated, while cell viability dropped to around 80% at 3, 10, and 30 μM before being drastically affected at 100 μM.

Hits were further investigated with additional secondary assays. To visualize that the drugs' effect on α-SMA stress fibre formation, immunocytochemistry was performed (Fig 3). As can be seen from Fig 3, stress fibre formation was reduced in the cells treated with 10 μM terfenadine, ebastine, or solifenacin.

To corroborate the findings, RT-qPCR and Western blot were performed. As can be seen in Fig 4, ACTA2 gene expression is significantly upregulated by TGF-ß1-treatment, which can be abrogated by treatment with 10 μM terfenadine, ebastine, or solifenacin (Fig 4A). This translates to protein level, as can be seen in Fig 4B. Western blot analysis shows that TGF-ß1-treatment leads to increase in α-SMA production, which is prevented in cells treated with 10 μM terfenadine, ebastine, or solifenacin.

Total ECM formation was quantified in another functional secondary assay (Fig 5). As can be seen in Fig 5, ECM deposition was reduced in the cells co-incubated with TGF-ß1 and the three drugs, compared to TGF-ß1 only. At 10 μM terfenadine, ebastine and solifenacin reduced ECM formation to levels of untreated fibroblasts.

Additionally, the effect of the drugs on the expression of a specific ECM protein (fibronectin) was evaluated on both mRNA and protein level. As evidenced by Fig 6A, TGF-ß1 treatment leads to an increase in FN2 gene expression, which is prevented by 10 μM terfenadine and ebastine, but not solifenacin. The same observation can be made on protein level. The Western blot in Fig 6B depicts bands with higher intensity at 250 kD for the TGF-β1 and TGF-β1 with 10 μM solifenacin groups, with less intense bands in the blank and other treatment groups.

To evaluate the effect of the drugs on cell proliferation, DRAQ5 nuclear staining was plotted after cells were treated in co-incubation of TGF-ß1 and the drugs with drug concentrations ranging from 0.03 to 100 μM. Fig 7 represents the concentration-dependent effects of the drugs on cell viability and therefore cell proliferation by proxy. Both solifenacin and ebastine only affected cell viability at the highest concentration tested (100 μM), while terfenadine treatment led to a decrease in cell viability to around 80% at 3, 10, and 30 μM before being drastically affected at 100 μM.

## Discussion

### Validity of the assay

The assay used for this screening campaign follows the 'phenotypic rule of 3', a rule for phenotypic assays, assessing relevance of the assay system, stimulus, and endpoint readout [29]. We

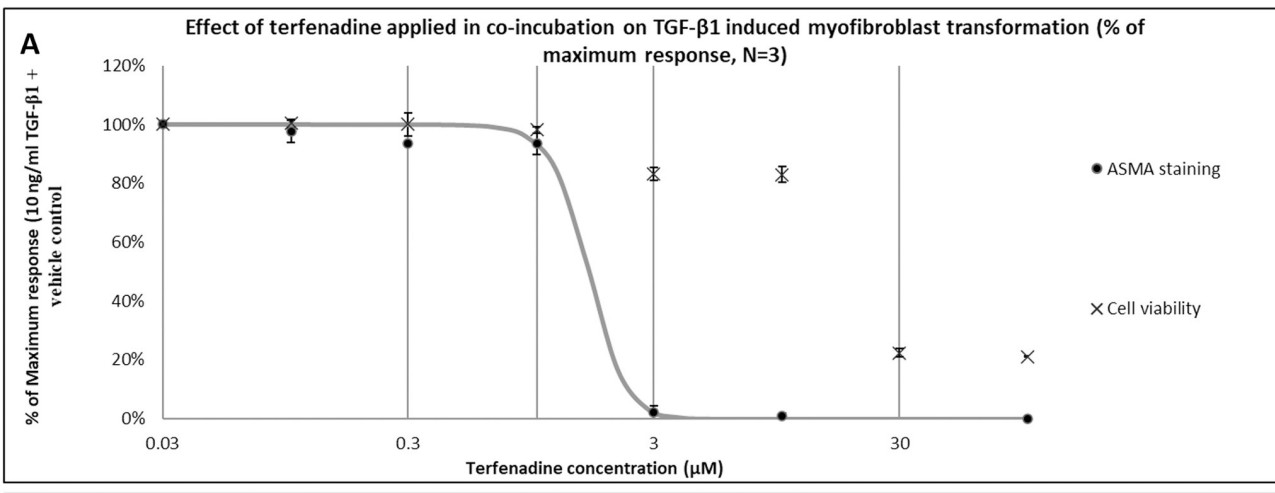

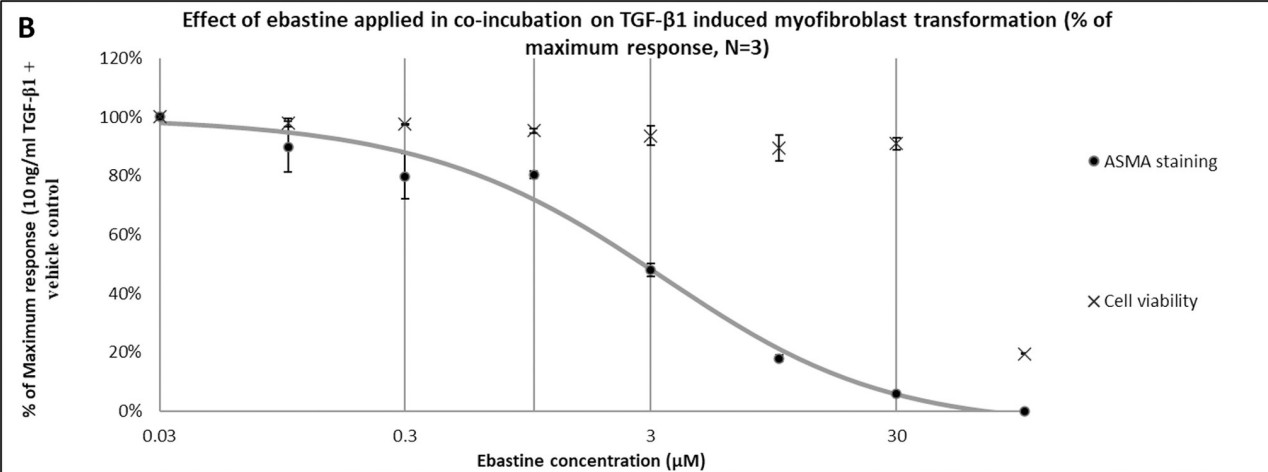

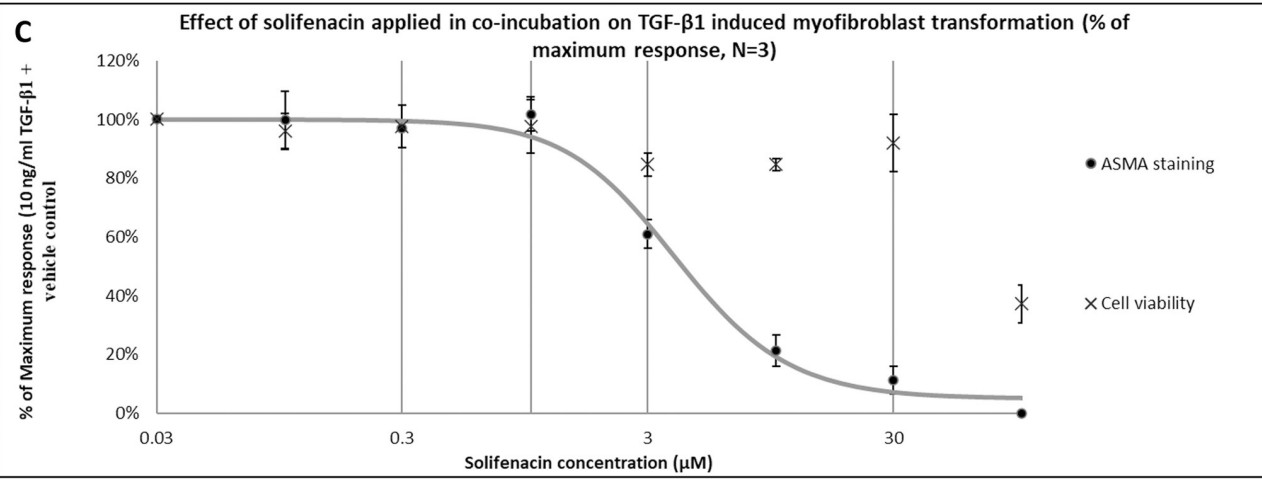

**Fig 2. Confirmation of concentration-dependent anti-fibrotic effect of hit drugs.** To ensure library integrity and the concentration-dependent anti-fibrotic effect of hit drugs, three hits were purchased from another supplier and tested on TA-derived fibroblasts. TA-derived cells were exposed to a range of concentrations of (**A**) terfenadine, (**B**) ebastine, or (**C**) solifenacin in co-incubation with 10 ng/mL TGF-ß1 for 72h. Cells were stained for α-SMA using ICE. Data points plotted as average ± SEM of the percentage of maximum response of α-SMA/DNA staining ratio (N = 3).

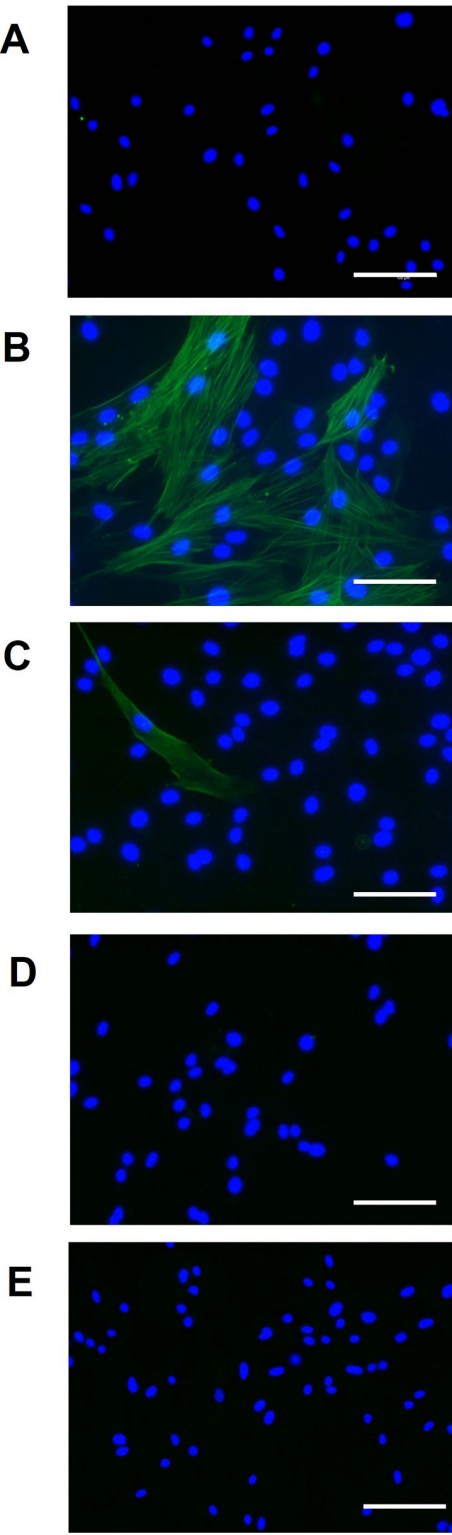

**Fig 3. Effect of drugs on α-SMA stress fibre formation.** To further confirm the anti-fibrotic effect of the drugs, immunocytochemistry was performed to visualize α-SMA stress fibre formation. Representative images shown for TA-derived cells that were left untreated (A), treated with 10 ng/ml TGF-ß1 (B), or in co-incubation with 10 ng/ml TGF-ß1 and 10 μM terfenadine (C), 10 ng/ml TGF-ß1 and 10 μM ebastine (D), or 10 ng/ml TGF-ß1 and 10 μM solifenacin (E). After 72 h of incubation, cells were stained for α-SMA expression (green). Nuclei were stained with DAPI (blue). White bars correspond to 100 μm. Images were captured at x200 magnification using a Leica DFC3000 G camera.

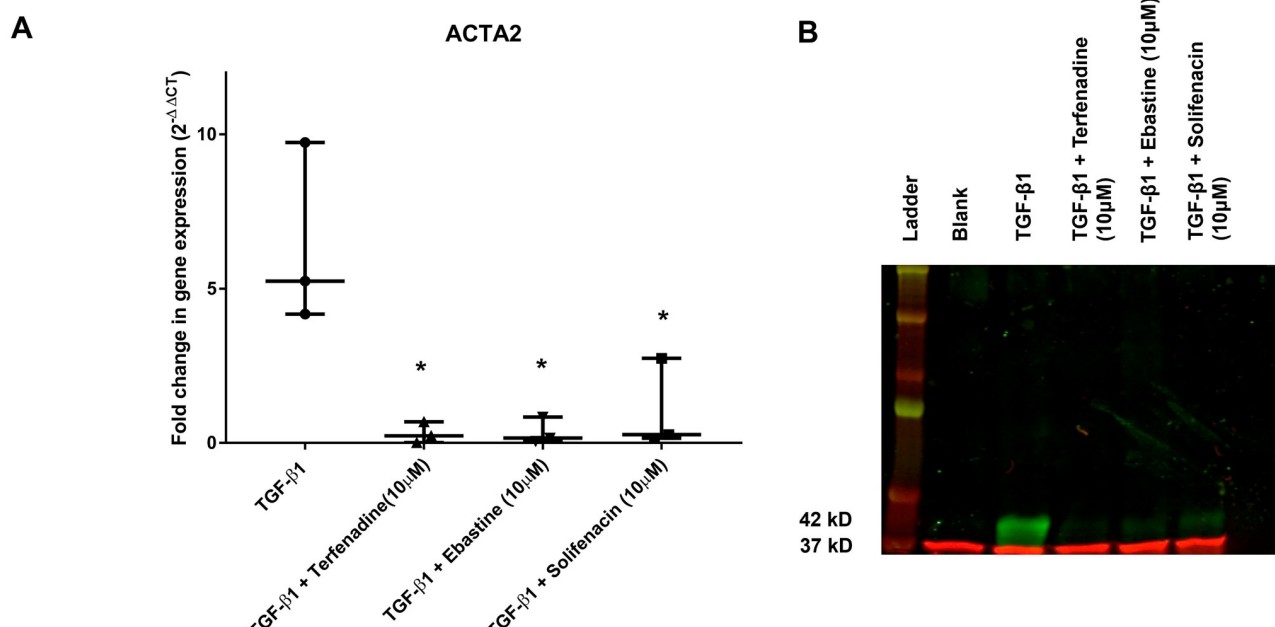

**Fig 4. Effect of drugs on α-SMA gene and protein expression.** To further confirm the anti-fibrotic effect of the drugs, RT-qPCR and Western blot was performed to quantify/visualize ACTA2/α-SMA gene and protein expression. TA-derived cells were left treated with 10 ng/ml TGF-ß1, untreated or in co-incubation with 10 ng/ml TGF-ß1 and 10 μM terfenadine, 10 ng/ml TGF-ß1 and 10 μM ebastine, or 10 ng/ml TGF-ß1 and 10 μM solifenacin. After 3 days of incubation, RNA or protein was isolated. Fold-change in ACTA2 gene expression (A) is plotted as box & whisker plot. Statistical analysis using one-way analysis of variance revealed that TGF-ß1 only group is significantly different to the other groups (p<0.05). Representative Western blot for α-SMA content in protein lysates of conditions outlined above (B). 20 μg of protein lysates was loaded under reducing conditions. Lower band in red (37 kD) denotes GAPDH loading control indicating equal loading, higher band at 42 kD represents α-SMA in green.

used primary human cells to increase the physiological relevance of the assay, as primary cells retain phenotypes seen *in vivo* compared with immortalised cell lines [36]. The critical role of TGF-β1 in PD has long been suggested [15, 37], and it is generally accepted that TGF-β1 is the main driver of myofibroblast transformation [18, 38, 39], making it an ideal candidate for a relevant stimulus in our screening assay.

Our proximal readout is the disease-linked protein biomarker α-SMA; *de novo* expression of this protein has been described as one of the key markers of myofibroblasts [21, 40]. Further, the relevance of α-SMA positive myofibroblasts in PD has been highlighted by others [10, 15]. Whilst the assay does not measure the ability of the drugs to inhibit TGF-β1 activation, it does quantify TGF- β1-induced myofibroblast transformation, a downstream consequence of TGF- β1 activation. When establishing the assay, multiple optimization steps were taken. We determined the optimum TGF-ß1 exposure time (24h, 48h, 72h) and optimum TGF-ß1 concentration (via concentration response curve ranging from 0.001 to 30 ng/mL). The optimum cell density was established, as well as the choice and concentration of primary antibody. It was found that $5 \times 10^3$ cells/well yielded the most consistent results with the greatest TGF-ß1-induced fold-change in α-SMA expression. The secondary antibody concentration was also carefully titrated. We further established the ideal fixation solution and duration (4% PFA for 20 min yielded the best results). We determined whether the use of serum free media would improve the TGF-β1-induced fold-change, as well as well-to-well consistency. We tested serum concentrations of 10%, 1%, and 0.1% and assessed the influence on TGF-β1-induced fold-change of α-SMA protein expression. No significant difference could be

## Effect of drugs on ECM production

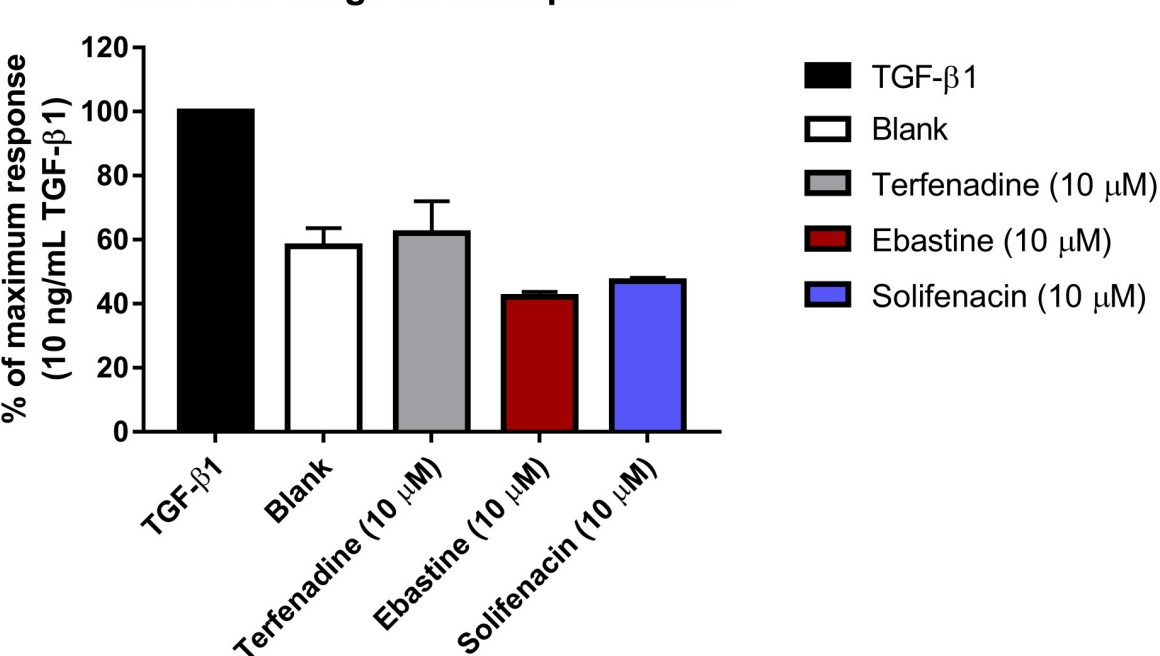

**Fig 5. Effect of drugs on ECM formation.** To further substantiate the anti-fibrotic effect of the drugs, a secondary assay quantifying TGF-ß1-induced ECM formation was performed. TA-derived cells were left treated with 10 ng/ml TGF-ß1 (black bar), untreated (white bar) or in co-incubation with 10 ng/ml TGF-ß1 and 10 μM terfenadine (grey bar), 10 ng/ml TGF-ß1 and 10 μM ebastine (red bar), or 10 ng/ml TGF-ß1 and 10 μM solifenacin (blue bar). After 7 days of incubation, cells were lysed, and total ECM was stained using Coomassie Blue. Data shown as average ± SEM of percentage of maximum response (TGF-ß1 control) (N = 3).

noted in fold-change, however, serum deprivation caused cell loss, leading to well-to-well inconsistency resulting in a less robust screening assay. Therefore, it was concluded that serum free conditions were not beneficial in our settings.

A battery of secondary assays that measured other aspects such as α-SMA stress fibre formation or ECM formation (two hallmarks of the myofibroblast phenotype), along with RT-qPCR and Western blot confirmed the hits that were followed up via concentration response curves, further substantiating the relevance of the primary screening assay. ICC visualised the α-SMA stress fibre formation, which was abrogated by treatment with 10 μM terfenadine, ebastine, or solifenacin. It should be noted that TGF-ß1-treatment increased the size of nuclei, which is in line with previous reports, characterising myofibroblasts as larger cells with more prominent Golgi apparatus and rough endoplasmic reticulum with elongated, serrated nuclei and extensive stress fibre network [41, 42].

### Hit rate

We have observed a hit rate of 1.3% which is in accordance with other similar phenotypic screening campaigns, such as a primary screen in fibroblasts to discover novel inhibitors of necroptosis (hit rate 1.4%) [43] and the 2.2% hit rate of a phenotypic screening campaign to identify novel antimalarial leads [44]. Another example of a similarly high hit rate is the 2,000 hits after screening a 200,000-compound library for compounds to modulate splicing of a gene relevant to spinal muscular atrophy in mice [45]. The hit rate of 1.3% is not uncommon for phenotypic screening campaigns [27].

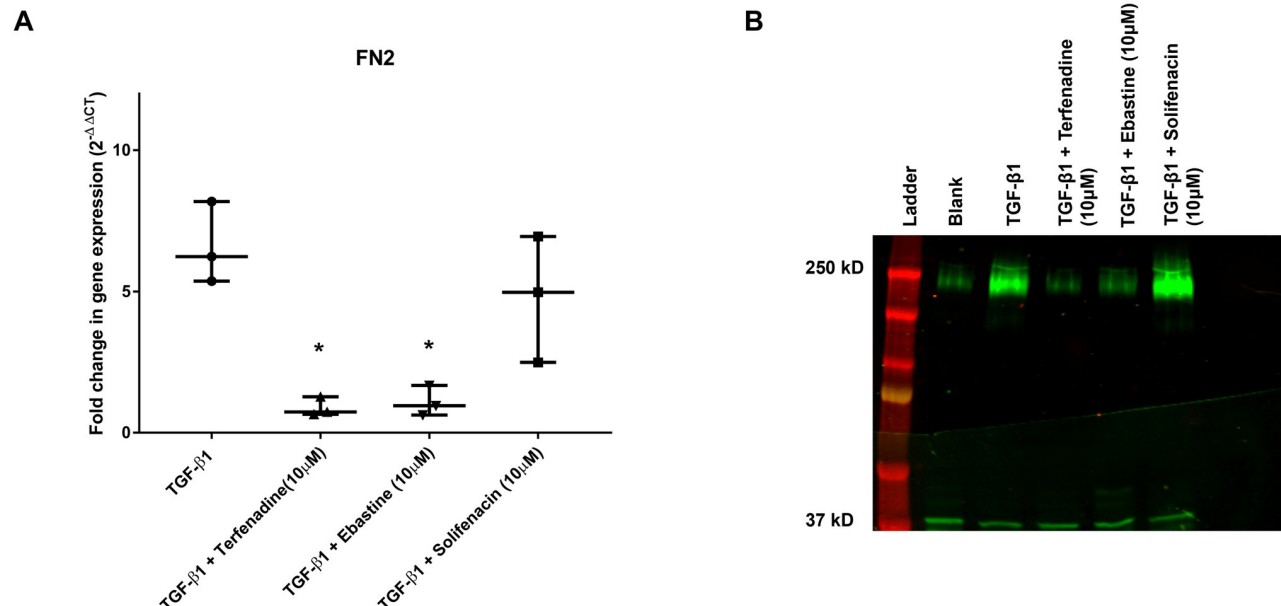

**Fig 6. Effect of drugs on fibronectin gene and protein expression.** To further confirm the anti-fibrotic effect of the drugs, RT-qPCR and Western blot was performed to quantify/visualize FN2/fibronectin gene/protein expression. TA-derived cells were left treated with 10 ng/ml TGF-ß1, untreated or in co-incubation with 10 ng/ml TGF-ß1 and 10 μM terfenadine, 10 ng/ml TGF-ß1 and 10 μM ebastine, or 10 ng/ml TGF-ß1 and 10 μM solifenacin. After 3 days of incubation, RNA or protein was isolated. Fold-change in FN2 gene expression (A) is plotted as box & whisker plot. Statistical analysis using one-way analysis of variance revealed that TGF-ß1 only group is significantly different to the groups treated with 10 ng/ml TGF-ß1 and 10 μM terfenadine or 10 ng/ml TGF-ß1 and 10 μM ebastine (p<0.05) but not the group treated with 10 ng/ml TGF-ß1 and 10 μM solifenacin. Representative Western blot for α-SMA content in protein lysates of conditions outlined above (B). 20 μg of protein lysates was loaded under reducing conditions. Lower band in green (37 kD) denotes GAPDH loading control indicating equal loading, higher band at 250 kD represents fibronectin in green.

## Review of the 26 hits

Ten of the 26 hits identified in this study were cancer drugs (kinase inhibitors, DNA damage related or targeting the cytoskeleton), which mostly have side effects that would not be

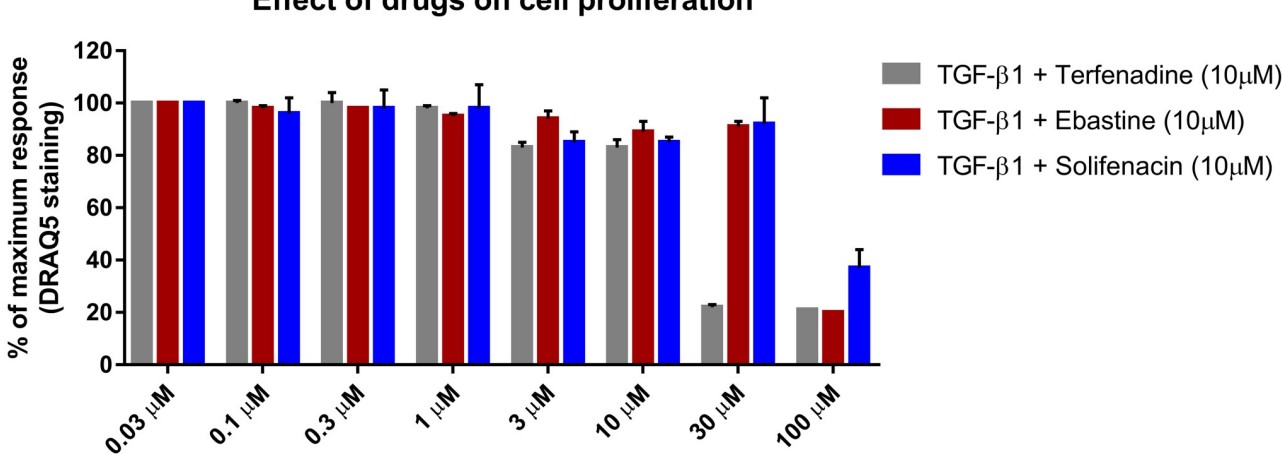

**Fig 7. Effect of drugs on cell proliferation.** To ensure drugs did not negatively affect cell proliferation, nuclear staining was quantified as a proxy for proliferation. TA-derived cells were exposed to a range of concentrations of terfenadine (grey), ebastine (red), or solifenacin (blue) in co-incubation with 10 ng/mL TGF-ß1 for 72 h. Cells were stained with DRAQ5 using ICE. Data points plotted as average ± SEM of the percentage of maximum response of DNA staining (N = 3).

desirable for an early-stage PD patient who typically presents with a penile nodule but is otherwise healthy. It is not surprising that these cancer drugs have inhibited myofibroblast transformation as it has been suggested that multi-kinase inhibitors with the ability to disrupt multiple signalling pathways could be a viable treatment for fibrosis by targeting multiple downstream mediators [10]. To our best knowledge, none of these drugs have been trialled for use in PD.

Histamine-related drugs were prominent among the hits, evidenced by histamine receptor 1 and 3 antagonists, histamine release blockers, and histamine N-methyl transferase inhibitors. Histamine receptor modulation has been suggested for hepatic and pulmonary fibrosis [46, 47]. The role of histamine release inhibition and histamine N-methyl transferase inhibition in the treatment of PD has not been described previously.

Muscarinic receptor antagonism (by drugs such as solifenacin succinate) has been utilised as a therapeutic mechanism in COPD [48], whilst, in contrast, it has been suggested that choline can attenuate cardiac fibrosis by agonist action on $M_3$ muscarinic receptors, regulating TGF-β1/Smad2/3 in *in vitro* and *in vivo* models of cardiac fibrosis [49]. Surprisingly, whilst total ECM production was reduced by solifenacin succinate treatment, fibronectin expression was at similar level to TGF-β1 treated cells. These results suggest that solifenacin succinate may be keeping the cells in the protomyofibroblast state, which is characterised by expression of fibronectin in absence of α-SMA stress fibres [50], as opposed to preventing the entire myofibroblast transformation process. To our knowledge, muscarinic receptor antagonism has not been investigated in PD.

Chlorotrianisene is a non-steroidal oestrogen and a precursor of SERMs which we have previously shown to have anti-fibrotic effects in *in vitro* and *in vivo* models of PD, whilst a clinical effect could be shown in the early stages of the disease [24, 32].

Molidustat is an inhibitor of HIF prolyl-hydroxylase and increases endogenous production of erythropoietin. It has been previously described to have anti-fibrotic effects in models of kidney disease *in vitro* or *in vivo* which has been attributed to suppression of myofibroblast transformation by HIF stabilization [51] but has not been tested in PD.

Aceclofenac, a COX inhibitor, has been observed to inhibit peridural fibrosis in lumbar laminectomies in rabbits [52]. COX signalling has been suggested to have a critical role in the development of liver fibrosis [53], whilst, in contrast, COX-2-derived prostacyclin has been suggested to prevent bleomycin-induced pulmonary fibrosis [54]. Chronic COX inhibition, however, has also been linked with increased intestinal fibrosis [55], potentially limiting the use of the drug for fibrosis. COX inhibition has not been investigated in PD.

Conflicting data has been published on the anti-fibrotic effect of dexamethasone in models of pulmonary fibrosis, where it has been suggested to work in a short term treatment after acute injury in rats [56], but not inhibit fibroproliferative effects in chronic patients [57]. This could be explained by the drug preventing rather than reversing myofibroblast transformation. Additionally, it has been suggested that combination therapy of dexamethasone and berberine might be more beneficial [58].

It has been previously suggested that dopamine receptor D2 antagonists, such as, prochlorperazine dimaleate, exert anti-fibrotic effects in models of diabetic hepatic fibrosis and non-alcoholic steatohepatitis induced liver fibrogenesis by regulating TGF-β1/Smad in hepatic stellate cells, and decreasing YAP levels in macrophages, respectively [59, 60]. To our knowledge this pathway has not been investigated in PD.

It is conceivable that at least one of the contrast agents (methylene blue) is a false positive, due to blue autofluorescence being detected by the plate reader, while another contrast agent (Gadopentetate dimeglumine) has been restricted by the EU due to evidence of gadolinium deposition in brain tissues [61].

Some drug classes such as statins had potential (mevastatin, fluvastatin, atorvastatin and lovastatin all eliciting 100% inhibition) but impacted cell viability negatively, despite the previously reported anti-fibrotic action of simvastatin in primary human fibroblasts derived from PD-patients by preventing nuclear translocation of YAP/TAZ [62]. Going forward, we will be constructing full concentration-response-curves with these drugs in combination with other drugs in order to further our understanding of their pharmacology in myofibroblast transformation.

## Drugs that increased myofibroblast transformation

Out of the drugs that were shown to increase myofibroblast transformation, five target VEGFR, which explains the pro-fibrotic effect given the proposed anti-fibrotic nature of VEGF [63]. The drugs with pro-fibrotic effect still have potential to be repurposed for other indications such as diabetic foot ulcers, open wounds that are slow to heal or other diseases where accelerated wound healing would be clinically beneficial but should be avoided for early-stage PD patients. Interestingly, nintedanib, one of two approved drugs for pulmonary fibrosis, was among the drugs that increased myofibroblast transformation the most which in contrast to a study in lung fibroblasts [64], further substantiating the claim for heterogeneity in fibrosis as discussed below.

## Drugs that increased cell viability

We have observed that some drugs increased the cell viability compared to the control conditions, which can be a sign of increased cell density and an increased level of metabolic activity, as proliferation depends on biosynthesis of macromolecules which are generated via multiple metabolic pathways [65, 66]. It has further been described that drug treatment can have an impact on growth rates and $GR_{50}$ values [67]. For example for B-RAF inhibitors, it has been described that in wild type (non-cancer mutation) B-RAF cells, the drugs paradoxically activate MAP kinase signalling and proliferation [68]. The TGFBRI inhibitor galunisertib has previously been described to reduce proliferation of fibroblasts in presence of TGF-β1 [69]. In that study however NIH3T3 cells, rather than primary human fibroblasts, were used. In addition, it has been reported in other cell types that complete abrogation of TGF-β1 signalling can lead to increased proliferation [70].

## Heterogeneity in fibrosis

Peyronie's disease is a fibrotic disease and fibrosis can affect any tissue or organ in the human body [25]. Although fibrotic diseases share a common pathophysiological mechanism which is centred around transformation of resident cells like fibroblasts to myofibroblasts leading to excessive production of ECM proteins by myofibroblasts, they seem to differ from each other in their responsiveness to treatment. For example, only two approved drugs exist for fibrotic diseases which are pirfenidone and nintedanib and only licenced for pulmonary fibrosis. Clinical data show that these two drugs are not effective in other fibrotic diseases [71–73], this has also been confirmed in our study, as these drugs inhibited myofibroblast transformation in our assay by -2% and -105% for pirfenidone and nintedanib respectively, indicating that they actually increased myofibroblast transformation. This suggests that there may be heterogeneity in fibrotic diseases; meaning that a drug identified for a fibrotic disease may not be efficacious for another fibrotic disease. Indeed, a study that used a previous, smaller version of the FDA-approved drug library to investigate potential anti-myofibroblast effect in commercial human lung fibroblasts, identified alprostadil as a hit [64]. In our study this drug only exhibited 25% inhibition at 10 μM. Another study in myelofibrosis assessed 150 FDA-approved drugs for

their ability to inhibit fibroblast proliferation but did not fully quantify myofibroblast transformation. While not entirely transferable, their data also suggests that the anti-fibrotic effect of the drugs they discovered is disease-specific [74]. Heterogeneity within fibrotic diseases in responsiveness to drugs may result from heterogeneity of fibroblasts; it has been suggested that even within the same organ there may be different fibroblast populations [75]. It has also been suggested that that developmental origin, anatomical site and location within the tissue architecture all play a role in fibroblast identity [76]. With fibroblast heterogeneity being reported even at tissue level for fibrotic lungs [77] and fibrotic skin disorders [78], it is not surprising to see heterogeneity between different tissues. Taken together, the implication of these observations would be that there is no 'silver bullet' that can be used to treat all fibrotic diseases.

## Limitations

This study is limited by the fact that not all the hits were followed up with full concentration-response-curves and that secondary assays were performed for only the drugs that were followed up. Therefore, some caveat must be added to the potential clinical translation of these results, as we do not know where the true $IC_{50}$ lies for most of the hits. This is important, as non-clinical studies often use concentrations that are much higher than the clinically relevant dose or even unachievable in patients, suggesting the need for concentration-response-curves that would inform the minimally effective clinical concentration [79]. The screening assay itself is limited by it being a monolayer, mono-culture model with a single cytokine insult, however this model was also used previously to discover a drug combination [24] which ultimately showed efficacy in patients [80].

With screening libraries as large as the one utilized in this project, it is only feasible to use one cell line derived from a single patient, which opens the possibility of genetic bias due to patient specific mutations. However, we believe that the tight error bars on the three hits that were followed up by full concentration response curves in different cells from other patients (N = 3) suggest this might not be an issue. Further, given the agnostic nature of a phenotypic assay, it is not possible to define a mechanism of action, without thorough target deconvolution.

The study is further limited by the choice of cell viability assessment. Several methods of assessing cell viability only capture a fraction of it (mitochondrial activity, metabolic activity, protease activity, membrane integrity) often overlapping with cell proliferation assays (DNA synthesis, metabolic activity). It is therefore necessary to use several assays to study the effects of compounds on these metrics in order to obtain the full picture. We endeavour to conduct these experiments in the future.

## Clinical translation

The drugs we identified in our previous screening campaign of 21 compounds using the same assay [24] have not been picked out as hits in this study, as we chose stricter criteria in the screening campaign presented here; where previously we just wanted to see effect and accepted $IC_{50}$ values between 3.5 and 30 μM. In the previous study, we identified PDE5i and SERMs as hits and demonstrated that the anti-fibrotic action of the combination of the two drug classes (PDE5is and SERMS) was synergistic which allows to lower the dosage of drugs. The combination of PDE5i and SERMs showed a synergistic anti-fibrotic effect in *in vitro* and *in vivo* models of PD [24] and the anti-fibrotic effect could be shown in early-stage PD patients [80]. We have retrospectively analysed the clinical effect of a combination of tamoxifen and PDE5i which were prescribed as off-label according to the UK regulations [81, 82]. The initial results show the combination treatment was able to slow the progression of the disease in patients

with early PD [80]. These results suggest that phenotypic assay of myofibroblast transformation is able to detect drugs that can be translated into clinic. Therefore, the further hits presented in this paper have the potential to be translatable to clinic.

## Potential impact on the fibrosis field

We chose to reveal all data from our screening campaign, and forego patenting, in an effort to help inform future repurposing efforts. Off-label use of drugs has a higher success rate if the rationale behind the drug use is grounded in scientific data, additionally the current dataset helps to avoid using drugs that would actually worsen the disease [83]. While not every hit that we identified will be viable to repurpose, we believe that the full dataset that is presented here will help advance the field, as it can provide a starting point for both novel treatments and novel basic science.

## Conclusions

Our screening campaign identified drugs that are approved for other indications with potential for repurposing for early-stage PD patients. Hits can either be repurposed directly or their targets can be used to develop novel chemical entities. Future prospective clinical trials should be conducted in the early, active phase of PD.

We hope that revealing the dataset in its entirety will lead to more informed drug repurposing efforts and help avoid drugs that have less potential benefit or are potentially harmful for PD patients.

## Supporting information

**S1 Table. Full list of 1,953 drugs.** The full list of 1,953 drugs along with their inhibition, cell viability, indication and other information, such as CAS number, target, DMSO (mg/mL) maximum solubility, DMSO (mM) maximum solubility, water (mg/mL) maximum solubility, water (mM) maximum solubility, URL, pathway, chemical formula, form, synonyms, SMILES (simplified molecular-input line-entry system), ALogP, and molecular weight.
(XLSX)

## Acknowledgments

The authors thank Stephen Bustin for designing primers for the RT-qPCR experiments.

## Author Contributions

**Conceptualization:** Marcus M. Ilg, David J. Ralph, Selim Cellek.

**Data curation:** Marcus M. Ilg, Alice R. Lapthorn.

**Formal analysis:** Marcus M. Ilg.

**Funding acquisition:** David J. Ralph, Selim Cellek.

**Investigation:** Marcus M. Ilg.

**Methodology:** Marcus M. Ilg.

**Project administration:** Marcus M. Ilg, Selim Cellek.

**Resources:** Selim Cellek.

**Supervision:** Selim Cellek.

**Validation:** Marcus M. Ilg, Alice R. Lapthorn.

**Writing – original draft:** Marcus M. Ilg.

**Writing – review & editing:** Marcus M. Ilg, Alice R. Lapthorn, David J. Ralph, Selim Cellek.

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
