## [Decision Letter · Decision Letter 0]

14 Mar 2022

PONE-D-22-05095Phenotypic screening of 1,953 FDA-approved drugs reveals 26 hits with potential for repurposing for Peyronie’s diseasePLOS ONE

Dear Dr. Cellek,

Thank you for submitting your manuscript to PLOS ONE. After careful consideration, we feel that it has merit but does not fully meet PLOS ONE’s publication criteria as it currently stands. Therefore, we invite you to submit a revised version of the manuscript that addresses the points raised during the review process.

We look forward to receiving your revised manuscript.

Kind regards,

Donald Gullberg, PhD

Academic Editor

PLOS ONE

Journal Requirements:

Additional Editor Comments:

An expert in the field has taken part of your work together with an editor.

They both find your work potentially interesting, and the reviewer has a number of suggestions for additional experiments that need to be performed.

We ask your to study the reviewer comments carefully and return a point by point response letter where your have addressed each of the issues raised in the review report.

Please let me know if you need additional time to do this.

I look forward to receiving your revised version.

Reviewers' comments:

Reviewer's Responses to Questions

**Comments to the Author**

1. Is the manuscript technically sound, and do the data support the conclusions?

Reviewer #1: Partly

2. Has the statistical analysis been performed appropriately and rigorously? 

Reviewer #1: I Don't Know

3. Have the authors made all data underlying the findings in their manuscript fully available?

Reviewer #1: No

4. Is the manuscript presented in an intelligible fashion and written in standard English?

Reviewer #1: Yes

5. Review Comments to the Author

Reviewer #1: The manuscript investigates whether phenotypic screening can be used to repurpose drugs for the treatment of Peyronie’s disease (PD). Authors have used primary fibroblasts from PD patients in a phenotypic screening assay that measures TGF-β1 induced αSMA protein synthesis. Using this screening assay, authors have screened 1,953 FDA-approved drugs and have identified 26 drugs that inhibits TGF-β1 induced αSMA protein synthesis. Moreover, the authors have also identified 10 drugs that increases TGF-β1 induced αSMA protein synthesis. The authors suggest phenotypic screening of FDA-approved drugs for PD is a viable method to predict drugs for repurposing to treat early PD. Although, the approach is interesting and supported with some experimental data, many issues remain to be addressed.

Major comments:

1. Secondary experiments like western blotting or qPCR or αSMA stress fibers by immunofluorescence for those 36 drugs that affected αSMA protein synthesis would be helpful to validate the phenotypic screening assay.

2. Apart from αSMA, other assays such as collagen gel contraction assay or TGF-β reporter assay using tMLC system would help to strengthen the data on myofibroblast transformation

3. Cell proliferation assay for those selected 36 drugs would be useful to evaluate the effects of these drugs

Minor comments:

1. The phenotypic screening assay measures the ability of the drugs to inhibit TGF-β1 induced αSMA protein synthesis but does not measures the ability of the drugs to inhibit TGF-β1 activation

2. In the results section, the isolated fibroblasts were mentioned as vimentin negative and needs to be clarified. In methods, section it is written as vimentin positive.

3. Low serum or serum-free conditions would have helped to reduce the effects from growth factors or other protein in the serum

6. PLOS authors have the option to publish the peer review history of their article (what does this mean?). If published, this will include your full peer review and any attached files.

Reviewer #1: No

---

## [Author Response · Author response to Decision Letter 0]

22 Jun 2022

Reviewer #1: The manuscript investigates whether phenotypic screening can be used to repurpose drugs for the treatment of Peyronie’s disease (PD). Authors have used primary fibroblasts from PD patients in a phenotypic screening assay that measures TGF-β1 induced αSMA protein synthesis. Using this screening assay, authors have screened 1,953 FDA-approved drugs and have identified 26 drugs that inhibits TGF-β1 induced αSMA protein synthesis. Moreover, the authors have also identified 10 drugs that increases TGF-β1 induced αSMA protein synthesis. The authors suggest phenotypic screening of FDA-approved drugs for PD is a viable method to predict drugs for repurposing to treat early PD. Although, the approach is interesting and supported with some experimental data, many issues remain to be addressed.

Major comments: 

1. Secondary experiments like western blotting or qPCR or αSMA stress fibers by immunofluorescence for those 36 drugs that affected αSMA protein synthesis would be helpful to validate the phenotypic screening assay.

We thank the reviewer for the suggestion of these secondary assays. We expanded the chapter ‘Confirmation of hits’ and have now included a figure that uses immunocytochemistry that demonstrates the effect of the selected three hit drugs on αSMA stress fibre formation, as suggested (Fig 3). We further tested these three drugs on ECM production and cell viability (please see the point below). We used these assays to confirm the three randomly selected hit drugs that we followed up via concentration-response-curves previously. Whilst it would be interesting to follow up all 26 hits that decreased myofibroblast transformation, it is beyond the scope of this manuscript which aims to report the findings from 1,953 drugs We have previously validated the assay after it revealed drugs in a smaller screening campaign and these hits (PDE5 inhibitors and selective estrogen receptor modulators) were validated using both, in vitro and in vivo experiments, highlighting the ability of the assay to identify relevant drugs (Ilg et al, 2019). Furthermore, the combination of these two drug classes have been shown to be effective in patients with early PD in the clinic (Megson et al 2022) which further validates the applicability of our screening assay. 

2. Apart from αSMA, other assays such as collagen gel contraction assay or TGF-β reporter assay using tMLC system would help to strengthen the data on myofibroblast transformation

We thank the reviewer for the suggestion and have included the outcome of an ECM production assay to further confirm the anti-myofibroblast activity of the three randomly selected hit drugs (Fig 4). As can be seen, ECM production is significantly reduced upon drug treatment, further confirming anti-myofibroblast activity of the drugs.

3. Cell proliferation assay for those selected 36 drugs would be useful to evaluate the effects of these drugs

Whilst we agree with the reviewer that it would be of interest to include data on cell proliferation, we only included the data for the three drugs that we previously confirmed via concentration response curve. We have now included a separate figure (Fig 5) to showcase that cell proliferation increases in cells treated with TGF- β1, which is reduced after treatment with the drugs. The graph depicts cells number (proliferation by proxy) derived from the nuclear stain DRAQ5 which emits at 700nm. 

Minor comments:

1. The phenotypic screening assay measures the ability of the drugs to inhibit TGF-β1 induced αSMA protein synthesis but does not measures the ability of the drugs to inhibit TGF-β1 activation

The reviewer is right to point this out. Whilst we cannot claim that the drugs prevent TGF- β1 activation, we can say that the drugs inhibit TGF- β1-induced myofibroblast transformation, a downstream consequence of TGF- β1 activation. Given the agnostic nature of a phenotypic assay, it is not possible to suggest a potential mechanism of action, without target deconvolution. We have now reworded the text where this was unclear. 

2. In the results section, the isolated fibroblasts were mentioned as vimentin negative and needs to be clarified. In methods, section it is written as vimentin positive.

We apologize for this mistake. The cells were desmin negative and vimentin positive. This has now been corrected in the results section. We thank the reviewer for pointing out this mistake.

3. Low serum or serum-free conditions would have helped to reduce the effects from growth factors or other protein in the serum

We agree with the reviewer on this, as this was our initial thought as well. When we developed the assay, we determined whether the use of serum free media would improve the TGF- β1-induced fold-change and/or well/well consistency. We tested serum concentrations of 10%, 1%, and 0.1% and assessed the influence on TGF- β1-induced fold-change of αSMA protein expression. No significant difference could be noted in fold-change, however, serum deprivation caused cell loss, leading to well/well inconsistency resulting in a less robust screening assay. Therefore, it was concluded that serum free conditions were not beneficial in our settings. We have now added a section in the discussion chapter to explain all optimisation steps we have taken while developing this assay.

---

## [Decision Letter · Decision Letter 1]

20 Jul 2022

PONE-D-22-05095R1Phenotypic screening of 1,953 FDA-approved drugs reveals 26 hits with potential for repurposing for Peyronie’s diseasePLOS ONE

Dear Dr. Cellek,

Thank you for submitting your manuscript to PLOS ONE. After careful consideration, we feel that it has merit but does not fully meet PLOS ONE’s publication criteria as it currently stands. Therefore, we invite you to submit a revised version of the manuscript that addresses the points raised during the review process.

We look forward to receiving your revised manuscript.

Kind regards,

Donald Gullberg, PhD

Academic Editor

PLOS ONE

Additional Editor Comments (if provided):

Thank you for the revised version which one editor and one reviewer have read.

Both think that data is interesting but reviewer has raised some serious concerns ( please see reviewer comments) and has proposed additional experiments and also ask to see raw data to clarify concerns.

Please study reviewer comments carefully.

I look forward to a revised version where you address the concerns raised point by point and also provide the raw data asked for. I  would be happy to give you extra time should this be needed.

Reviewers' comments:

Reviewer's Responses to Questions

**Comments to the Author**

1. If the authors have adequately addressed your comments raised in a previous round of review and you feel that this manuscript is now acceptable for publication, you may indicate that here to bypass the “Comments to the Author” section, enter your conflict of interest statement in the “Confidential to Editor” section, and submit your "Accept" recommendation.

Reviewer #1: (No Response)

2. Is the manuscript technically sound, and do the data support the conclusions?

Reviewer #1: Partly

3. Has the statistical analysis been performed appropriately and rigorously? 

Reviewer #1: Yes

4. Have the authors made all data underlying the findings in their manuscript fully available?

Reviewer #1: Yes

5. Is the manuscript presented in an intelligible fashion and written in standard English?

Reviewer #1: Yes

6. Review Comments to the Author

Reviewer #1: The authors have made considerable efforts to address some of the issues mentioned in the last review. However, there are still some major concerns that needs to be addressed.

Major comments:

1. The resolution of the images in figure 3 could have been better and quantification was not presented. Quantification of αSMA normalized to F-actin (Phalloidin) would make the data solid. Alternatively, the authors should consider either western blotting or qPCR for the three hit drugs

2. The ECM production assay to validate the myofibroblast function is appreciated but it would have been much better if the authors have performed either western blotting or qPCR for Fibronectin or Collagen I/III as these methods are not dependent on cell numbers

Minor comments:

1. According to the description in the figure legend, the images were captured at 200x magnification but scale bar was not presented. The nucleus size appear different across the images. Raw images of these data would help to evaluate the data better.

2. The amount of ECM measured using this assay can be influenced by cell proliferation and washing steps with ammonium hydroxide.

3. It is very hard to find a difference between the data presented in Figure 2 and figure 5 with reference to cell proliferation and cell viability as it uses the same method. The authors should have used an alternative method to assess cell proliferation.

7. PLOS authors have the option to publish the peer review history of their article (what does this mean?). If published, this will include your full peer review and any attached files.

Reviewer #1: No

---

## [Author Response · Author response to Decision Letter 1]

19 Oct 2022

6. Review Comments to the Author

Reviewer #1: The authors have made considerable efforts to address some of the issues mentioned in the last review. However, there are still some major concerns that needs to be addressed.

Major comments:

1. The resolution of the images in figure 3 could have been better and quantification was not presented. Quantification of αSMA normalized to F-actin (Phalloidin) would make the data solid. Alternatively, the authors should consider either western blotting or qPCR for the three hit drugs

>>>We thank the reviewer for the suggestion and have now included both RT-qPCR data for ACTA2 as well as Western blot images for ASMA to support the findings (new Fig 4). As can be seen in the figure, they concur with our previously reported observations.

2. The ECM production assay to validate the myofibroblast function is appreciated but it would have been much better if the authors have performed either western blotting or qPCR for Fibronectin or Collagen I/III as these methods are not dependent on cell numbers

>>>We have now included both RT-qPCR data for FN2 and Western blot images for fibronectin to support our findings (new Fig 6). The initial ECM production assay data is still included to show the effect of the drugs on total ECM formation, without differentiating the type of matrix protein. We have included a paragraph in the discussion to explain the effects of solifenacin succinate which we believe keeps the cells in the protomyofibroblast state instead of completely preventing myofibroblast transformation.

Minor comments:

1. According to the description in the figure legend, the images were captured at 200x magnification but scale bar was not presented. The nucleus size appear different across the images. Raw images of these data would help to evaluate the data better.

>>>We thank the reviewer for the comment. We have now included scale bars to make comparison easier. It should be noted that it has been reported that cellular size is affected in myofibroblast transformation. Myofibroblasts have been described as larger cells with more prominent Golgi apparatus and rough endoplasmic reticulum with elongated, serrated nuclei and extensive stress fibre network (PMID: 21297493, PMID: 32673537). This has now been added to the discussion. 

2. The amount of ECM measured using this assay can be influenced by cell proliferation and washing steps with ammonium hydroxide.

>>>We agree with the reviewer and have now included Western blot images for fibronectin as suggested (new Fig 6). It might not have been evident from the manuscript. but data was normalised to cell count before lysis. We have revised the methods to reflect this better (Line 217).

3. It is very hard to find a difference between the data presented in Figure 2 and figure 5 with reference to cell proliferation and cell viability as it uses the same method. The authors should have used an alternative method to assess cell proliferation.

>>>We do agree with the reviewer that cell proliferation assay would have been nice to show one of the characteristics of transformation of fibroblasts to myofibroblasts. However, several methods of assessing cell viability only capture a fraction of it (mitochondrial activity, metabolic activity, protease activity, membrane integrity) often overlapping with cell proliferation assays (DNA synthesis, metabolic activity). In order to measure cell proliferation adequately, several different assays have to deployed. We believe this would be outside the scope of our manuscript. We have now highlighted this as a point for future research in the limitations section.

---

## [Decision Letter · Decision Letter 2]

2 Nov 2022

Phenotypic screening of 1,953 FDA-approved drugs reveals 26 hits with potential for repurposing for Peyronie’s disease

PONE-D-22-05095R2

Dear Dr. Cellek,

We’re pleased to inform you that your manuscript has been judged scientifically suitable for publication and will be formally accepted for publication once it meets all outstanding technical requirements.

Kind regards,

Donald Gullberg, PhD

Academic Editor

PLOS ONE

Additional Editor Comments (optional):

Reviewers' comments:

Reviewer's Responses to Questions

**Comments to the Author**

1. If the authors have adequately addressed your comments raised in a previous round of review and you feel that this manuscript is now acceptable for publication, you may indicate that here to bypass the “Comments to the Author” section, enter your conflict of interest statement in the “Confidential to Editor” section, and submit your "Accept" recommendation.

Reviewer #1: All comments have been addressed

2. Is the manuscript technically sound, and do the data support the conclusions?

Reviewer #1: Yes

3. Has the statistical analysis been performed appropriately and rigorously? 

Reviewer #1: Yes

4. Have the authors made all data underlying the findings in their manuscript fully available?

Reviewer #1: Yes

5. Is the manuscript presented in an intelligible fashion and written in standard English?

Reviewer #1: Yes

6. Review Comments to the Author

Reviewer #1: The authors have addressed the questions raised in the last review with new experiments.

Minor comment: In figure 6A, fibronectin gene is denoted as FN2 instead of FN1

7. PLOS authors have the option to publish the peer review history of their article (what does this mean?). If published, this will include your full peer review and any attached files.

Reviewer #1: No

---

## [Editor Report · Acceptance letter]

1 Dec 2022

PONE-D-22-05095R2 

Phenotypic screening of 1,953 FDA-approved drugs reveals 26 hits with potential for repurposing for Peyronie’s disease 

Dear Dr. Cellek:

I'm pleased to inform you that your manuscript has been deemed suitable for publication in PLOS ONE. Congratulations! Your manuscript is now with our production department. 

Kind regards, 

on behalf of

Professor Donald Gullberg 

Academic Editor

PLOS ONE